# OpenReview forum: "Phare: A Safety Probe for Large Language Models"
_ICLR.cc/2026/Conference — ICLR 2026 Conference Withdrawn Submission_

### Official Review · Reviewer_FL6D · 2025-11-01

**Soundness:** 3
**Presentation:** 2
**Contribution:** 2
**Rating:** 4
**Confidence:** 3

**Summary:**

This paper introduces PHARE, a multilingual diagnostic framework designed to systematically evaluate the safety of large language models (LLMs) across three dimensions: hallucination & reliability, social biases & stereotypes, harmful content generation. It evaluates 17 leading LLMs, identifying recurring safety vulnerabilities such as sycophancy, prompt sensitivity, and bias reproduction.

**Strengths:**

1. The paper introduces a novel diagnostic benchmark that focuses on identifying how and why LLMs fail on safety dimensions, rather than just ranking models.
2. The study offers practical insights showing that prompt style, user confidence, and system brevity instructions directly influence factual reliability.
3. It is methodologically comprehensive and reproducible, releasing data and code across hallucination, bias, and harmfulness evaluations.

**Weaknesses:**

1. **Lack of Coherent Design Across the Three Dimensions**

The primary concern is that the benchmark’s three dimensions (bias & fairness, hallucination, harmfulness) appear to be designed and evaluated independently, without a unifying methodological framework. As a result, the benchmark feels more like a concatenation of three separate datasets rather than a single, well-integrated evaluation suite. A clearer central design philosophy or shared task structure would strengthen the contribution.

2. **No Unified Scoring Metric Across Dimensions**

Related to the previous point, the benchmark does not provide an overall score or aggregated evaluation metric for each model. This makes it difficult to compare models holistically or to understand their overall safety capability. The paper would benefit from a unified scoring scheme (e.g., weighted or normalized composite score) that enables direct cross-dimension model comparison.

3. **Lack of Discussion on How to Improve Safety Capabilities**

The paper thoroughly evaluates safety shortcomings of current models but provides limited insight into potential mitigation strategies

4. **Writing and Structure Issues**

- The order of the three dimensions in the introduction differs from the ordering in Sections 2 and 3 (Intro: Bias → Hallucination → Harmfulness; Sections: Hallucination → Bias → Harmfulness). Consider maintaining a consistent order throughout the paper.

- Section 4 is titled Perspectives—why not follow convention and name it Conclusion instead?

**Questions:**

NA

---

### Official Review · Reviewer_CYGZ · 2025-11-05

**Soundness:** 2
**Presentation:** 2
**Contribution:** 2
**Rating:** 4
**Confidence:** 2

**Summary:**

The paper presents Phare, a multilingual diagnostic framework for probing LLM safety across three axes: (i) hallucination/reliability, (ii) social bias and stereotypes, and (iii) harmful content generation. The benchmark emphasizes culturally grounded prompts curated from diverse, non-English sources and evaluates models across multiple languages (e.g., English, French, Spanish). Empirical analyses highlight systematic vulnerabilities such as sycophancy, prompt sensitivity, and stereotype reproduction, offering a unified lens for cross-model comparison.

**Strengths:**

- The writing and figures are clear and easy to follow.
- Unlike many English-centric benchmarks (e.g., those derived primarily from Wikipedia), Phare intentionally draws from culture-specific sources to diversify prompts.
- Safety is assessed across multiple dimensions within one framework.

**Weaknesses:**

- Experiments center on English, French, and Spanish—three Indo-European languages—providing narrow evidence for “multilingual” claims. Typologically distant and low-resource languages are not represented, limiting generalizability.
- The manuscript emphasizes multilinguality (Table 1 and narrative) but omits discussion and comparisons to MultiJail [1], a multilingual safety benchmark covering 10 languages. This omission weakens the novelty and positioning.
- Although culturally grounded prompts are highlighted as a key feature, the results do not provide disaggregated analyses by culture/region/community or demonstrate concrete multicultural insights (e.g., where and why models fail across cultures).
- The safety aspects (hallucination, bias/stereotypes, harmful content) are already covered in prior work. Without stronger evidence of new measurement methodology or new insights, the contribution risks appearing primarily as a repackaging of existing dimensions.

[1] Deng, Yue, Wenxuan Zhang, Sinno Jialin Pan, and Lidong Bing. "Multilingual jailbreak challenges in large language models.", ICLR, 2024

**Questions:**

- The proposed self-coherency score relies on LLM-as-judge, but no validation against human annotations, inter-rater agreement, or robustness checks is reported. How well does the score correlate with human judgments?

---

### Official Review · Reviewer_aqet · 2025-11-06

**Soundness:** 3
**Presentation:** 3
**Contribution:** 2
**Rating:** 4
**Confidence:** 4

**Summary:**

This paper introduces Phare, a multilingual diagnostic framework for evaluating large language models (LLMs) across three safety dimensions: (1) hallucination and reliability, (2) social biases and stereotypes, and (3) harmful misguidance. Phare combines culturally specific prompt generation in English, French, and Spanish, statistical association measures (Cramér’s V) for bias quantification, a self-coherency probe comparing generative and discriminative awareness of stereotypes, and a structured tool-use evaluation to detect parameter hallucinations.

Empirically, Phare finds that confident user tone reduces debunking accuracy, concise output instructions lower factual robustness, tool-use accuracy deteriorates under missing or extra parameters, and harmful misguidance is comparatively well mitigated. The framework is multilingual, transparent, and statistically rigorous—though many of its findings confirm well-documented behaviors such as sycophancy and verbosity effects rather than introducing new conceptual insights.

Disclosure: I previously reviewed an earlier version of this paper during the NeurIPS 2025 Datasets & Benchmarks cycle and participated in the full author–reviewer discussion. I have evaluated the current submission independently, but I note this prior involvement.

**Strengths:**

(1) Comprehensive and transparent framework. \
Phare integrates hallucination, bias, and harmful-content evaluations into a single, modular pipeline with open data, clearly defined prompts, and reproducible scoring methods. The framework is well-documented and replicable, setting a strong standard for benchmark transparency.

(2) Rigorous statistical testing and validation. \
The analysis incorporates confidence intervals, significance testing with FDR correction, and human verification of LLM-judge reliability. The consistency between human and automated evaluations lends credibility to the results and demonstrates careful experimental design.

(3) Multilingual and culturally grounded dataset. \
Phare’s prompt generation draws from culturally specific materials in English, French, and Spanish rather than simple translations. This design increases ecological validity and provides useful insight into cross-lingual model behavior.

(4) Well-executed synthesis of known effects. \
While the key findings—such as reduced pushback under confident tone, factual degradation under brevity, and greater resilience in harmfulness tasks—are not particularly novel, their systematic and statistically grounded presentation across models and languages makes the benchmark a useful comparative tool for the field.

**Weaknesses:**

(1) Language aggregation and performance variation. \
It is unclear whether headline results are averaged across English, French, and Spanish or reported for English only. Aggregation can obscure meaningful cross-lingual differences, and the appendix suggests noticeable performance variation across languages, with English often strongest and other languages sometimes reversing rankings. Clarifying how results are aggregated and briefly discussing these cross-lingual trends would make the findings easier to interpret.

(2) Incremental conceptual novelty. \
The framework is executed with care but largely confirms established patterns: confident user tone reduces pushback, brevity degrades factual accuracy, and underspecified inputs impair tool reliability. These findings align with existing work on sycophancy, verbosity, and robustness. I appreciate the extensive evaluation across a wide range of state-of-the-art models, which provides a valuable comparative view of current systems. That said, Claude’s consistently strong safety performance is not particularly surprising, given the model family’s well-known emphasis on alignment and safety-finetuning.

(3) Unrealistic perturbations in the tool reliability evaluation. \
The synthetic omissions, additions, and conversions are systematic but somewhat artificial. Because the direction of the effect is predictable, these tests mainly show that imperfect inputs yield imperfect outputs rather than revealing deeper robustness properties. Including a small set of human-authored, realistic perturbations—for instance, typographical errors, vague or incomplete referents, or ambiguous user instructions—would provide a more naturalistic test of model reliability.

(4) Bias module: construct validity and baselines. \
The association-matrix approach intentionally avoids baseline classes, but prior approaches such as Marked Personas (Cheng et al., 2023, ACL) and CrowS-Pairs (Nangia et al., 2020, EMNLP) rely on explicit “marked vs. unmarked” comparisons (e.g., woman vs. man, Black vs. White, in-group vs. out-group) because many social harms are relative to an in-group baseline. Measuring only absolute co-occurrence frequencies risks overlooking disparities that emerge as differences from these reference groups. The absence of such baseline contrasts limits the ability to capture these relative or asymmetric biases.

(5) Cardinality, aggregation, and hidden heterogeneity. \
The paper would benefit from clarifying how Cramér’s V associations are aggregated across attribute classes and how this process handles within-attribute heterogeneity. Attributes with many discrete values can yield large, sparse contingency tables in which most category pairs have weak or zero counts. When the resulting $\Chi^2$ statistics are averaged over all combinations, these many near-zero entries can depress the mean, making the overall association appear small even if certain category pairs are strongly correlated. Moreover, aggregation can conceal asymmetric or localized stereotypes: for instance, if only one group within an attribute (e.g., “Muslim”) is systematically associated with negative professions while others are not, the overall religion × profession association may still appear weak. In such cases, low global association values may reflect averaging across neutral in-group categories rather than genuine absence of bias. Clarifying exactly how the authors compute and aggregate Cramér’s V—whether per attribute pair or globally—and discussing how the method handles high-cardinality and asymmetric distributions would make the results more interpretable.

(6) Limitation of attribute-only associations. \
Many salient stereotypes concern qualities rather than protected categories—for example, “mothers are more caring than fathers,” as discussed in Stereotyping Norwegian Salmon (Abid et al., 2021, ACL). Because Phare’s analysis focuses on associations among demographic attributes, it cannot capture such trait-based stereotypes that relate to perceived qualities or competencies. It would be helpful for the authors to discuss this design choice and its implications, since much of the prior fairness and bias literature focuses on these quality-based associations. Contextualizing this limitation would clarify how the framework’s scope compares to existing bias-evaluation methods

(7) Diagnostic rather than normative meaning of the self-coherency metric. \
The self-coherency probe provides a valuable diagnostic signal but does not directly measure safety at generation time. A model with high association and high self-coherency recognizes that its generated patterns are stereotypical yet still reproduces them—indicating post-hoc awareness of bias without behavioral correction. Conversely, low self-coherency could arise either because the model fails to identify a harmful stereotype or because it correctly treats a benign, real-world correlation as acceptable. In both cases, the metric captures recognition–generation consistency, not ethical soundness.

Moreover, because the "debatable vs. acceptable pattern" judgment is itself made by an LLM, the classification reflects the model’s internal normative assumptions rather than an external ground truth. Interpreting this as a direct measure of safety would therefore be misleading: high self-coherency simply means the model is internally consistent, not that it behaves safely. It would strengthen the paper to emphasize this diagnostic interpretation explicitly and to report association rates (what the model generates) and recognition rates (what it later identifies as acceptable or biased) separately, along with their difference as a measure of internal alignment. Finally, including a small human-rated subset would help calibrate these LLM judgments against human annotations and provide an external validity check.

**Questions:**

1. Could you clarify whether the headline results are averaged across English, French, and Spanish or reported for English only? The appendix suggests notable performance differences across languages; it would help to briefly comment on how aggregation was handled and whether any consistent cross-lingual patterns emerged.

2. For the bias module, could you clarify how Cramér’s V is aggregated across attribute pairs? Is it computed separately for each attribute pair and then summarized, or aggregated across all category combinations within a pair? This distinction affects how low global association values should be interpreted, particularly when attributes have many categories or asymmetric distributions.

3. The framework deliberately avoids marked vs. unmarked contrasts (e.g., woman vs. man, Black vs. White, in-group vs. out-group) that are standard in earlier bias evaluations such as Marked Personas and CrowS-Pairs. Many historically salient stereotypes are defined precisely through contrast with a baseline or dominant group—for instance, perceptions of “difference” from a presumed norm rather than independent association strength. Could you elaborate on the reasoning behind this design choice? Is the goal primarily to simplify and automate measurement, or do you view the absolute-association approach as conceptually preferable despite potentially missing these contrastive forms of bias?

4. Based on the expansive prior literature in the bias/stereotype metric space, Phare’s reliance on absolute attribute-association measures and the self-coherency probe represents departure from how bias has often traditionally been evaluated in LLMs. However, the conceptual motivation for this design is not especially well justified or convincing. Could you elaborate on why this formulation was chosen over more established comparative methods, and how you see it advancing our understanding of bias beyond diagnostic convenience?

5. The self-coherency probe measures internal consistency between a model’s generative behavior and its subsequent recognition of stereotypes. Philosophically, though, awareness of bias does not necessarily equate to behavioral safety, since a model might still reproduce the same associations even when it recognizes them as problematic. Could you clarify whether self-coherency is intended to capture ethical awareness, behavioral safety, or internal diagnostic consistency, and how that interpretation connects to your broader framing of "safety" in Phare?

---

### Note · Authors · 2025-11-20

I have read and agree with the venue's withdrawal policy on behalf of myself and my co-authors.